# Risk Factors for Eating Disorders and Perception of Body in Young Adults Associated with Sex

**DOI:** 10.3390/nu13082819

**Published:** 2021-08-17

**Authors:** Aleksandra Purkiewicz, Anna Malwina Kamelska-Sadowska, Joanna Ciborska, Julia Mikulska, Renata Pietrzak-Fiećko

**Affiliations:** 1Department of Commodity Sciences and Food Research, Faculty of Food Science, University of Warmia and Mazury in Olsztyn, Plac Cieszyński 1, 10-719 Olsztyn, Poland; aleksandra.purkiewicz@uwm.edu.pl (A.P.); renap@uwm.edu.pl (R.P.-F.); 2Department of Rehabilitation and Orthopedics, School of Medicine, Collegium Medicum, University of Warmia and Mazury in Olsztyn, 10-719 Olsztyn, Poland; 3Clinic of Rehabilitation, Provincial Specialist Children’s Hospital in Olsztyn, 10-719 Olsztyn, Poland; 4Department of Human Nutrition, Faculty of Food Science, University of Warmia and Mazury in Olsztyn, Słoneczna 45 F, 10-719 Olsztyn, Poland; joanna.ciborska@uwm.edu.pl; 5Cosmetology, Trichology, Dietetics Julia Mikulska, 11-500 Giżycko, Poland; mikulskajulia@wp.pl

**Keywords:** food, eating habits, consumption, sex, binge eating, anorexia nervosa, compulsive overeating, eating disorders

## Abstract

(1) Background: The integrated approach to the prevention and treatment of eating disorders (EDs) requires knowledge and can be used only when specific risk factors are known. The aim of this study was to examine the differences in food choices and eating behavior between males and females; (2) Methods: This study comprised 148 females and 27 males aged from 18 to 26-years-old (MEAN ± SD = 21.4 ± 1.86 years old). Information about EDs was obtained from four different measures: the body mass index (BMI), the eating disorder screen for primary care (ESP), a standardized and validated questionnaire called “My Eating Habits” and the food frequency questionnaire with 10 answers (FFQ-10); (3) Results: The risk for developing eating disorders was detected in nearly 67% of respondents. It was also shown that EDs were more common in females and how body weight affected the way individuals feel about themselves. Females showed more unhealthy eating habits, which contributed to dietary restrictions and emotional overeating, as they were also afraid of gaining weight. The frequency of eating meat and drinking alcohol was higher in males, whereas eating legume seeds was less frequent in females. (4) Conclusions: This study opens a new field, which will help health care professionals recognize the problems with eating disorders and treat them based on different sex characteristics.

## 1. Introduction

Eating disorders (EDs) are common psychiatric disorders with an increasing number of medical and social problems particularly associated with high mortality [1]. An integrated approach to prevention and treatment for EDs requires knowledge and can be used only when specific risk factors are known [2]. The early detection of bad eating habits that pose a risk for the development of an ED is the most important step in decreasing the spread of EDs [3]).

The growing prevalence of EDs is caused by stressful conditions created by beauty images shown in media or by the improper amount or quality of food distributed to consumers. Both eating excessively, as well as not enough food, could contribute to health problems [4], such as anorexia, binge-eating disorder (BED), bulimia and obesity [5,6]. In addition, psychological factors closely related to the way in which people engage in eating behavior play an important role in the development of EDs [2].

Eating disorders are caused by a variety of factors that are unclear. Many experts, on the other hand, feel that a mix of genetic, physical, social, and psychological variables may play a role in the onset of an eating disorder. Eating disorders can also be influenced by social pressure. In Western culture, physical attractiveness and a thin physique are frequently associated with success and personal value. Eating disorder behaviors may be fuelled by a desire to succeed or be accepted. A study showed a reduction in eating disorder risk factors among female adolescents due to parental intervention. The intervention showed a reduction in body shaming and an increase in healthy eating habits. Without parental support, an increase in negative body image and binge eating was observed [7]. Negative body feelings and reduced body protection fully mediated the relationship between self-esteem and binge eating, regardless of gender [8].

The susceptibility of different risk factors largely depends on sex and age [9]. The incidence of EDs is significantly higher among females compared to males [10]. The latter is explained by the following multifactorial determinants: (1) physiological factors related to adolescence and a greater increase in adipose tissue among girls during this period and (2) psychological factors related to females sharing a greater need to compete and focus on their look and sexual appearance. Girls have a tendency to delay the puberty process by refusing to eat [11].

Previous studies show that males tend to overeat, whereas females are more likely to opt for loss of control while eating [10]. This and other research show that contrary to stereotypes, EDs are not just a problem among females. It is more difficult for males to talk about the problems in the nutrition sphere because they consider it “not masculine” and humiliating [12]. 

Boys in adolescence and adult males account for about 10% of ED diagnoses. The most common cases are bulimia, nervosa, compulsive overeating syndrome and obesity, while anorexia is relatively rare among males. Although some males are more exposed to bigorexia than young females [13,14]. 

Recent studies suggest a dynamic systems approach in the treatment of EDs, which is preferred. Any dietary and lifestyle change as profound as recovery from EDs will be crucial only when a full range of pragmatic (psychological, cultural, social, etc.) factors are involved [15]. 

The identification of risk factors in different sexes, as well as the diagnosis of ED itself, could allow health care professionals recognize and treat problems with severe underweight or obesity. 

Eating disorders are a multi-factorial problem, and this study showed how different factors (nutritional and non-nutritional) affect the prevalence and frequency of EDs occurrence. Based on the identification of behaviors characteristic of people suffering from eating disorders, including obsessive thinking about one’s appearance, feelings accompanying and after eating, disturbed body image, as well as typical behaviors and abnormal eating patterns, specialists diagnosing eating disorders have a broader point of view on the patient.

To date, there is no comprehensive study that has investigated the problem of eating disorders based on the analysis of three different questionnaires—ESP, MEH and FFQ. 

Additionally, the development of civiliation made the problem of eating disorders valid. This complementary research enabled us to show these issues in a holistic way, including the psychology of nutrition—ESP and MEH, as well as eating habits’ “relation” to food and dietary patterns, and FFQ are all used in this way. Additionally, eating disorder symptoms were assessed in a mixed gender community, and the influence of sex on three different measures were shown in this study. Based on the literature search and thus, the lack of knowledge in this field, research on specific risk factors in association with different sexes is needed. The aim of this study was to examine the differences in food choices and eating behavior between males and females. Information gained from this study could allow for the identification of health-related problems among the studied group of young people and allow for the tailoring of health education programs to address these problems in a holistic way.

## 2. Material and Methods

### 2.1. Participants

This study comprised 148 females and 27 males aged from 18 to 26 years old (MEAN ± SD = 21.4 ± 1.86 years old; Table 1). The results of the anthropometric measurements were shown in Table 1. Participants were assigned using an alphabetic code. All of the participants were white Caucasians. 

### 2.2. The Inclusion and Exclusion Criteria

The inclusion criteria for both groups were calendar age over 18 years (but less than 30 years), a sedentary lifestyle [confirmed by the International Physical Activity Questionnaire (IPAQ)], no medical contraindications to participating in the research study and the voluntary notification of the participant. 

The exclusion criteria for both groups included: the coexistence of other disabilities and diseases with the exception of overweight and obesity, any medications that may affect the results of the analysis, as well as those who were unable to read Polish or give valid consent. The exclusion criteria were also: acute or chronic diseases, such as heart problems, diabetes, asthma, inflammation, trauma, recent fractures of the bone, recent surgery—in the last 6 months), low-fat diet or a lipid-lowering agent. Moreover, the participants who did not feel well at the time of recruitment were excluded from the analysis. 

Participants were instructed to maintain their accustomed dietary and physical activity habits throughout the study participation period.

### 2.3. Ethics

All of the procedures performed in this study, involving human participants, conformed to the ethical guidelines of the 2013 Declaration of Helsinki, as reflected in a priori approval by the institution’s human research committee and followed the Adapted Physical Activity (APA) Ethics Standard. The protocol was approved by the Institutional Review Board (IRB) and ethics committee (20/2021). All participants provided written informed consent, and verbal assent regarding their participation in the study. Approval was obtained from an institutional research ethics committee prior to recruitment, and subjects received no payment for participation in this study. 

The experiment was conducted with the understanding of each participant. The anthropometric measurements as well as questionnaires were performed on separate days. The questionnaires were performed both verbally as well as via Internet. It was shown previously that using clinician verbal screening is better in the assessment of patient with food inconsistency [16]. The electronic questionnaire was created in the form of a website using http://gsuite.google.pl. The created link for a survey was placed on one of the social networks and dedicated for people aged 18–26.

The questionnaires were filled at the University of Warmia and Mazury in Olsztyn among students of the Faculty of Food Sciences (Food Technology and Human Nutrition and Gastronomy—culinary arts) and among students of Dietetics at the Faculty of Health Sciences. 120 questionnaires were obtained via the Internet, while verbal screening was performed in 65 people. 

### 2.4. Methods

The information about EDs, eating habits, food choice and motives, as well as feeding practices known as eating behavior, were obtained from four different validated measures: the body mass index (BMI), the eating disorder screen for primary care (ESP) (Table 2) [17]; a standardized and validated questionnaire called “My Eating Habits” by Ogińska-Bulik and Putyński [18] and the food frequency questionnaire with 10 answers (FFQ-10) [9].

#### 2.4.1. Anthropometrics and Body Mass Index (BMI) Calculations

Body mass (after the removal of shoes and heavy clothing) was measured to the nearest 0.1 kg. Body height was measured to the nearest 0.1 cm using a patient weighing scale with a height rod (Seca 217; Seca Poland, Medical Measuring Systems and Scales). Body mass index (BMI; kg m^−2^) was calculated as weight (in kg) ÷ height^2^ (in m^2^) to one decimal place.

#### 2.4.2. The Eating Disorder Screen for Primary Care (ESP)

The ESP questionnaire is shown in Table 2, and it was designed as a brief screening assessment to identify primary care patients who were at risk for EDs and in need of more specialized care [17]. EDs were diagnosed when participants answered at least two questions in a way indicating a problem with eating. 

#### 2.4.3. “My Eating Habits” Questionnaire

A “My Eating Habits” (MEH) questionnaire was standardized and validated on a normal and overweight sample by Ogińska-Bulik and Putyński [18]. The reliability of the questionnaire was assessed by internal consistency (Cronbach’s alpha), and it was high—0.89 [15]. It consisted of 68 statements, with YES or NO possible answers. The questions were classified into three categories: a. habitual overeating (HO) when performing certain activities; b. emotional overeating (EO), or eating excessive amounts of food under the influence of negative or positive emotions; c. dietary restrictions (DR) related to limiting the amount of food consumed [18]. The questionnaire was used to diagnose eating disorders and predict the tendency to gain weight. Each “Yes” answer was scored one (1) [except for five (5) appropriately marked questions]. The total score was used to calculate the mean and standard deviation for each category for all respondents and separately for sex groups.

#### 2.4.4. The Food Frequency Questionnaire (FFQ-10)

The food frequency questionnaire (FFQ) was shown to be valid and reliable to assess dietary intake in previous studies [19,20]. Participants were asked to indicate the frequency of intake of 62 groups of food items from 10 food groups over the past 12-month period. Clear instructions and pictures of common household measures and food portion photographs for each food item were provided. In addition to the 108 food items, the participants had the option to fill in a section named ‘Others’ in the items column, if there were any foods or beverages that did not appear on the FFQ food list. In order to estimate the intake frequency, 10 categories were used: (1) never, (2) once a month or less frequently, (3) 2–3 times a month, (4) 1–2 times a month, (5) 3–4 times a week, (6) 5–6 times a week, (7) 1 time a day, (8) 2–3 times a day, (9) 4–5 times a day and (10) 6 times a day or more [9].

### 2.5. Statistical Analyses

Statistical analysis was performed using Statistica 13.3 software (Statsoft Inc., Tulsa, OH, USA). All data were reported as mean ± SD, minimum and maximum values unless stated otherwise. 

The aim of the statistical analysis was to obtain information on the specific factors increasing the risk for EDs in males and females. The percentages and the exact number of people characterized by a given trait were used. The obtained values were rounded to decimal places.

The Pearson Chi-square test was used to assess the differentiation of the responses for questions between both groups of people. The results of the My Eating Habits questionnaire were computed in points, and because the distribution of the variables in the compared samples were sometimes different than normal distribution (Shapiro–Wilk test results), the Kruskal–Wallis test and Dunn test (post hoc) were used for testing the differences of average values between three classes of eating habits in male and female and the Mann–Whitney test for the comparison of the results between two sex group in each class of eating habits. Statistical significance was assumed at *p* < 0.05.

## 3. Results

The risk for EDs was detected in nearly 67% of respondents.

### 3.1. Anthropometrics and Body Mass Index (BMI)

Most of the respondents (63.4%) had a normal body weight (55.6% of males and 64.9% of females; Table 3). Being overweight was more common in males (males vs. females: 37.0% vs. 14.2%), whereas only females were obese (4.7%, *n* = 7). In males, only two of them had mild or moderate thinness, while 16% of females had mild thinness.

### 3.2. The Eating Disorder Screen for Primary Care (ESP)

The results of the ESP questionnaire are shown in Table 4. The ESP questionnaire is used to detect people at risk of developing EDs. Thus, inappropriate eating habits and an initial diagnosis of EDs were possible to perform. A total of 56% of the respondents were not satisfied with their eating patterns (males vs. females: 29.6% vs. 60.8%, respectively; *p* = 0.0027 Table 4.). 

Nearly 30% of respondents admitted eating in secret (males vs. females: 14.8% vs. 32.4%; *p* = 0.0027). In 76% of respondents, body weight influences the way they think about themselves. Almost 19% of participants admitted the presence of EDs in their family, and 20% of respondents suffer or suffered with them in the past. 

In conclusion, 66.3% of studied population was diagnosed as being at risk for EDs (males vs. females: 52% and 69%, respectively, *p* = 0.0849). 

Females were more prone to acquire eating disorders. On the other hand, the odds ratio (OR) and relative risk (RR) for the male population were: OR for men and the “Yes” response—0.486 (95% confidence interval OR = 0.211–1.115) and RR—0.752 (95% confidence interval RR = 0.515–1.099), respectively.

### 3.3. “My Eating Habits” Questionnaire Admit

The behaviors associated with habitual overeating are shown in Table 5. A large number of females (36.5%) admitted that they rarely overeat compared to males (11.1%; *p* = 0.0097;) and seldom feel to have overeaten (females vs. males; 45.3% vs. 22.2%; *p* = 0.0255; see Table 5). 

Table 6 and Table 7 show the results of the habits associated with emotional overeating.

A larger number of females (51.4%) felt guilty after eating too much food compared to males (14.8%; *p* = 0.0005). Females also admitted that they are not able to tell themselves “enough” during a meal (females vs. males; 30.4% vs. 11.1%; *p* = 0.0388; see Table 6).

Overeating anxiety was more common in female participants, as well as mood-dependent eating (65.5%; see Table 7).

The habits associated with dietary restrictions are shown in Table 8 and Table 9. The risk for dietary restrictions was shown by the results of three statements: (1) I pay too much attention to my weight; (2) I rarely feel guilty after overeating; (3) I sometimes avoid eating, even when I feel hungry (Table 9). It was shown that females (42.6%) pay too much attention to their body weight in comparison with males (22.2%; *p* = 0.0467; see Table 8). More than a half of the studied female population (50.7%) felt guilty after consuming too much food, whereas this problem affected about 22% of males respondents. 

Avoiding food even when feeling hungry affected more than 26% of females and 11% of males (Table 9).

In the group of males, there was no significant differentiation in the average value between the different classes of eating habits (Kruskal–Wallis test: H_(2, *N* = 81)_ = 2.822; *p* = 0.244); however, such a differentiation was noted between all three classes of eating habits in the group of females (Kruskal–Wallis test: H_(2, *N* = 444)_ = 26.211; *p* < 0.0001; Dunn test: dietary restriction vs. habitual overeating—*p* = 0.0258; dietary restriction vs. emotional overeating—*p* = 0.000001; habitual overeating vs. emotional overeating *p* = 0.042). Only in the emotional overeating class differences were found in the average test value between males and females (Table 10). 

In this study, the relation between previous or current EDs (e.g., anorexia nervosa) and “My Eating Habits” results in males and females were also analyzed (Table 11 and Table 12). It was shown that, in the overall population, nearly 86% who suffer or have ever suffered in the past from EDs were not satisfied with their body image (more females than males; see Table 11 and Table 12). Almost 86% of people who suffer or have suffered from EDs preferred to prepare their own meals. The desire to eliminate excess calories after a meal was admitted by over 71% of the entire population who suffer or have suffered from EDs. Emotional eating was expressed by over 36% of people who have never experienced EDs.

The behaviors typical for binge-eating disorder (BED) were also assessed in accordance with the occurrence of EDs now or previously (Table 13). More than half of those who suffer or have suffered from an EDs often eat without a large appetite (51.4% for the entire studied population). However, sex had significant impact on this value (56.3%; see Table 14). Approximately 54% of respondents admitted a problem with EDs and that they sometimes “stuff themselves” with food. People with present or past EDs (80% of this population group) admitted feeling guilty after a larger meal (this was typical for the female population; see Table 14).

This study showed the relationship between selected eating frequency behaviors and satisfaction with one’s eating habits (Table 15 and Table 16). More than half of the individuals who participated in the study who frequently skipped meals were not satisfied with their diet. This was typical for the female population (57.8%; see Table 16). Most of the people (60.6%) who were satisfied with their diet admitted that meal times were strictly defined. On the other hand, people dissatisfied with their diet (males vs. females 62.5% and 90.0%, respectively; see Table 15 and Table 16) rarely ate meals at strictly defined time frames. The conducted analysis showed that people who were satisfied with their diet less often skipped meals and ate more than three meals a day.

### 3.4. The Food Frequency Questionnaire (FFQ-10)

#### 3.4.1. The Consumption of Fruits and Vegetables

The results of FFQ showed that males and females were alike in some ways (Table 17). They all ate fruits and vegetables once a day. However, there were more females who declared eating vegetables 5–6 times a day, in comparison with males (males vs. females: 17.6% vs. 7.4%; see Table 17).

#### 3.4.2. The Consumption of Pulses

This study also showed the relationship between sex and the frequency of the consumption of dry, fresh and canned pulses (Table 17). The majority of females have never eaten any pulses. Similar results were obtained for males and females for frequencies of more than 1–2 times a week.

#### 3.4.3. The Consumption of Meat and Fish

Most males consumed red meat 3–5 times a week (25.9%; Table 17). Females consumed red meat once a month or less (22.3% of respondents). Almost 19% of males admitted that they ate red meat every day with a frequency of 1–2 times a week. Over 21% of females ate red meat 2–3 times a month, and 21% admitted that they did not eat red meat at all (for males 11.0%).

In the case of poultry, males ate it 3–5 times a week or once a day. Among nearly 19% of males, poultry was consumed 5–6 times a week, and nearly 15% ate it 2–3 times a day. Over 7% of both males and females consumed poultry meat once a month or less. Thus, the conducted analysis showed that males consumed red meat and poultry more frequently than females (*p* < 0.05).

More females than males ate lean fish with the frequency of 2–3 times a week (22.5% vs. 3.7%). However, males declared that they ate lean fish more often (3–5 times a week) (Table 17). Fatty fish were consumed 1–2 times a week more often by males than females (29.6% vs. 15.5%).

#### 3.4.4. The Consumption of Dairy Products

In this study, the consumption frequency of dairy products depending on sex was assessed (Table 17). The highest percentage of males consumed milk and natural milk drinks every day (22.2%), while nearly 4% did not consume these products at all. Most of the surveyed males consumed natural milk and milk drinks 2–3 times a day (18.5% of the entire studied group) or 1–2 times a week (18.5%).

Males consumed cheese once a day (25.9%). Almost 24% of females indicated eating cheese 3–5 times a week. Cheese was eaten least frequently by females with a frequency of 4–5 times a day (2% of the surveyed females) and with a frequency of 6 times or more a day (2% of surveyed females).

#### 3.4.5. The Consumption of Alcohol

Males more often consumed beer 5–6 times a week (22.2% of respondents), while a large percentage of females stated that they consumed beer 2–3 times a month (27.7%). About 11% of males consumed beer once a month or less frequently, while females consumed twice as much (24.3%). Males more often consumed strong liquor with a frequency of 2–3 times a month (25.9% of the group), once a month or less frequently or never (approximately 22% of respondents) (Table 17).

## 4. Discussion

Satisfying physiological needs, including hunger and thirst, has become a problem on a global scale nowadays. Despite widespread food availability, people’s diet is not proper enough to keep them healthy. 

It was shown previously that eating disorders (EDs) most often develop in adolescence and just after age 20 [21]. The results of this research in males and females aged 18–26 showed that a risk of EDs is present in nearly 67% of the population studied. 

This study showed that males and females are different in many issues concerning EDs. An important element of this study is that EDs cannot be limited solely to females, as 52% of males were diagnosed as being at risk for EDs. The diagnosis of EDs was obvious in case of females, because they put much emphasis on a lean physique. On the other hand, it was shown previously that males constitute a smaller proportion of people suffering from EDs, but this does not exclude them completely. Males less often declare episodes of starvation or obsessive weight control, which makes these abnormalities difficult to notice [22].

### 4.1. Anthropometrics and Body Mass Index (BMI)

This study showed that there are a lot of young people, both males and females, who were overweight (especially in the male population), as well as malnourished (especially in the female population) (see Table 3). Previous research in male youths 18 to 20 years old showed that mean BMI was within normal range (mean BMI was 20.5 among 20-year-old youths). However, the prevalence of underweight was 21.6% (higher than in this study), while the prevalence of overweight and obesity were 4.6% and 0.6%, respectively (lower than in this study) [23]. It was previously shown that BMI over 22 kg/m^2^ is a risk factor for diabetes later in life [24].

Małecka-Tendera et al. [25] showed that, over the last 30 years, the number of obese children and adolescents in the United States has increased sharply. The latter concerned also the European countries, including Poland. Data from 2008 show that approximately 60% of Poles over 20 were overweight and over 25% of them were obese. The prevalence of overweight was higher in men, but obesity was more frequent in women [22]. In the Polish population aged 20 to 74 years, BMI below 25 kg/m^2^ was found only in 47% of respondents [26,27].

### 4.2. The Eating Disorder Screen for Primary Care (ESP)

This study showed that females were more prone to develop EDs because of dissatisfaction with their eating patterns (see Table 4). More females than males were also not satisfied with their eating patterns in previous research (30.51% vs. 18.85%) [28]. A new study showed that female student-athletes had higher rates for risk of EDs than males, as shown from the ESP questionnaire (45.95% ± 3.03 vs. 13.66% ± 1.80, respectively) [29,30]. Other researchers also showed similar results—higher rates of EDs in females in comparison with males, both within the general population and amongst athletes [31,32]. A higher risk of EDs was also shown in a previous study [33].

Eating in secret is an eating behavior that leads to binge-eating disorder. The sick thus hide from others the amount of food they eat and the way in which it is eaten [27,28]. In this study, more females than males admitted eating in secret. This was consistent with previous studies. More females than males admitted eating in secret, family history and the influence of body weight on their perception [28].

### 4.3. “My Eating Habits” Questionnaire

In this study, higher anxiety in females was observed after eating too much food. This was similar in girls diagnosed in previous studies (females vs. males: 41.3% vs. 16.1%) [34]. The same as in the previous study, secret snacking was more common in females participants. More females also admitted to eating despite satiety [34] (compare with Table 5). Results obtained from this study, as well as previous research, showed that emotional overeating is most common in the females/girls population e.g., post-primary school students [34]. This research revealed that eating was a mood enhancer in females, same as in post-primary school students. A previous study also showed that girls were more likely show dissatisfaction with their body weight [34]. This study showed that DRs were not so common in the studied population. On the other hand, previous studies on eating habits in secondary school students showed similar values between all three categories (HO = 3.74 ± 2.49; EO = 3.46 ± 2.23; DR = 3.34 ± 2.15).

Previous research on dietary restrictions in females showed that 37% of them followed a restriction diet at least once. Almost all participants knew the guidelines and rules of particular types of different dietary programs [35].

Young females decide to follow restrictive diets because they want to achieve a slim, media-promoted figure that has become the equivalent of beauty nowadays. They put too much importance on their body image in order to feel more attractive to themselves and others [35]. This research showed that females in comparison with males felt guiltier after eating excessively (*p* ≤ 0.05).

This study showed the relationship between sex and individual behavior related to body weight. More than half of males (55.6%) and 58.1% females rarely controlled their body weight. Almost 76% of women considered themselves prone to gaining weight. It was shown previously that females more often suffer from psychopathological symptoms related to improper perception of their own body and obsessive fear of gaining weight [36]. Constant thoughts about body weight make the figure the subject of evaluation. These are the basic risk factors for EDs, especially anorexia nervosa [37]. Low self-esteem and constant criticism of one’s own body image may also cause overeating [38]. The greater preoccupation with body weight and the fear of gaining weight confirm that females are in the zone of a higher risk of developing an eating disorder. 

This study showed that almost 86% presented with behaviors typical of anorexics and orthorexics e.g., preparing meals themselves. People suffering from orthorexia meticulously choose their menu and plan to prepare their meals, even several days in advance [39]. On the other hand, people with anorexic features have unusual eating habits, such as cutting food into small pieces and counting bites while eating food, which makes them constantly preoccupied with thoughts about food [40]. Almost 63% of people who have had an ED in the past or present showed that food had too much importance to them. When analyzing behaviors typical of people with BED syndrome, a statistically significant occurrence of them were noted in people suffering from present or past EDs (*p* ≤ 0.05). More than half of the respondents (51.4%) ate without much appetite, and 54.3% of people experienced “stuffing”/overeating food. Additionally, 80% of people with current or past problems of EDs tend to experience remorse after overeating.

Similar findings were presented by Kobos et al. [41] with behaviors associated with habitual overeating and dietary restrictions declared by M = 3.66 and M = 3.59 secondary school students. As reported by the authors, emotional overeating was more common compared to our study (M = 4.84 and M = 3.46. respectively). Different results compared with our study were obtained by Ogińska-Bulik and Putyński [18] with dominant emotional overeating (M = 4.67) and less common habitual overeating (M = 2.94). In opposition to this research, in which the most answers were recorded in the group of emotional overeating questions (4.60), Janota et al. [34] showed responses from the habitual overeating group dominated (3.74). Both in the other authors’ research and authors’ of this study, respondents showed the least behaviors from the group of dietary restrictions. 

### 4.4. The Food Frequency Questionnaire (FFQ-10)

The research by Doubova et al. [42] proved that the frequency of consumption of vegetables and fruit is determined by sex—females eat them more often than males. The results of this research showed that fruits were eaten more frequently by males and vegetables by females. Sweet fruit preserves were consumed by the largest group of people 2–3 times a month (26.9% of respondents), while dried fruit was never included in the diet of over 40% of respondents. Food products, such as jams, plum syrup, fruit syrups and dried fruits, were classified as calorie-rich. They contain a high amount of sugar and are less recommended in the daily diet compared to fresh fruit [43]. The decreased consumption of these products in studied population could be considered a proper nutritional behavior.

The frequency of consumption of red meat differed depending on sex—18.5% of males ate it every day (for women—2%). Also, Bingham et al. [44] showed that men consume meat more often in comparison with females. Previous research by Love and Sulikowski [45] showed that males were statistically more likely to eat red meat. Taking into account the sex of the surveyed people, there were no statistically significant differences in the frequency of consumption of poultry meat among the group of people studied. Kubberod et al. [46] confirmed the hypothesis that red meat is more preferred by males. Moreover, this research showed that the sensory features of white meat, including poultry, are more preferred by females. Love and Sulikowski [45] studied the attitudes of males and females towards meat. There was a stronger correlation in males between the association of “meat” and “health” and between “meat” and “delicious”, which may be one of the reasons why males consume meat more frequently. The sex difference was not related to explicit attitudes to meat, nor was it attributable to a variety of other factors, such as a generally more positive disposition toward meat in males than females. Males also exhibited bias toward meats, compared to non-meat foods, while females exhibited more caution when searching for non-meat foods, compared to meat. These biases were not related to implicit attitudes but did tend to increase with increasing hunger levels [45]. 

Milk and milk products are a source of easily digestible calcium and phosphorus, which are bone-building components; therefore, it is recommended to consume these products twice a day [47]. About 22% of males and about 18% of females consumed milk and natural milk drinks every day. Additionally, 18.5% of males and about 11% of females consumed them with greater frequency, i.e., 2–3 times a day. Regarding cheese, almost 26% of males consumed cheese daily. More frequent consumption of milk and milk drinks among males was noted. Similar results in their research on the frequency of consumption of food products by students were obtained by Galiński et al. [48], who stated that milk and milk drinks were consumed more often by males. Taljić and Delalić [49], examining the relationship between the consumption of milk and dairy products in relation to sex, showed that more males consume milk and yogurt every day. Moreover, the authors’ research showed that males consume milk, yogurt and cheese more frequently than females. In turn, Kardas et al. [50], analyzing the frequency of the consumption of selected food products by males and females, found that sex has no influence on the frequency of consumption of milk and dairy products. 

Fish is a product with a high content of wholesome protein, polyunsaturated fatty acids from the omega-3 family and is a source of valuable minerals, including magnesium, calcium, iodine and selenium. Moreover, they are a source of B vitamins and in the case of fatty fish, vitamins A and D. Due to their pro-health values, they should be eaten at least twice a week [51]. A higher percentage of males (41%) than females (20.4%) consumed lean fish in accordance with the dietary recommendations, which was statistically significant. Moreover, 11% declared that they consumed lean fish with a higher frequency—3–5 times a week, while the percentage of females who consumed lean fish with this frequency was twice as low (5.4%). Almost two times more males (29.6%) than females (15.5%) ate oily fish 1–2 times a week. Can et al. [52] reports that females consume more fish per year than males. An inverse relationship was observed by Nanri et al. [53], which stated that males ate more fish than females. In the study group, more frequent consumption of alcoholic beverages was observed among males. Over 22% of males consumed beer 5–6 times a week, while the highest percentage of females (27.7%) consumed it 2–3 times a month. Strong liquor was consumed by almost 15% of males 5–6 times a week.

Similar results to this study were demonstrated by Ceylan-Isik et al. [54], which proved that males more often consumed alcoholic beverages. It is assumed that males consume greater amounts of alcohol, especially on special occasions, while females consistently abstain from alcohol consumption. However, the relationship is somewhat unclear, and it has been shown that an increasing number of females experience problems with excessive alcohol consumption [55]. Recent studies showed that beer consumption had increased among males, while wine consumption had increased among females. Moreover, females became addicted to alcohol more often [56].

It was shown that alcohol and eating disorders often go together. The problems with eating increased the risk of alcohol use disorder, and conversely, people who were addicted to alcohol were at risk of developing an EDs. Alcohol was commonly presumed to disinhibit food consumption. Giving up meals and restricting an individual’s diet to little or nothing allowed drinking larger amount of alcoholic beverages (binge-drinking) without fear of gaining weight [57]. Recent studies show 30 percent of women between 18 and 23 have skipped a meal in order to drink more [58]. Burke et al. [57] emphasized in their research that people suffering from EDs tended to consume too much alcohol more often.

Other research showed that alcohol use disorders frequently co-occurred with eating disorders, as alcohol was often used as for emotional regulation (if tension, anxiety, constant thoughts about food and appearance occur) or as a part of impulsive behavior in those suffering from unhealthy eating conditions [59]. It was also shown that disordered eating and alcohol use contribute to the unhealthy phenomenon of drunkorexia, which refers to condition of binge-drinking combined with the typical self-imposed starvation seen in anorexia nervosa. It also refers to individuals who use purging (as seen with bulimia nervosa) in order to reduce caloric intake to offset the calories consumed in alcohol.

People suffering from eating disorders turn to alcohol to feel relieved and tame their emotions. They could also easily become malnourished, while additionally suffering from all of the health and social risks associated with alcoholism. Therefore, the latter eating disorders often require therapy related to eating behavior, as well as alcohol dependence.

Furthermore, the role of disordered eating attitudes in the relationship between drunkorexia and emotion regulation, as well as emotion regulation difficulties, has been investigated [60]. Gender differences in adolescents have also been highlighted [59]. Drunkorexia was most prevalent among college-age females, although men and older ages of both genders have also been known to develop the disorder [61].

What is important in this case, the tendency to drink excessive amounts of alcohol may be inherited, which confirms the influence of genetic factors on the shaping of EDs [62].

Although there are many studies concerning the well-known harmful effects of alcohol abuse, there are new epidemiological findings showing positive consequences on human health e.g., a lower cardiovascular risk or a beneficial effect of beer on endothelial function [63]. Research on the impact of beer consumption on the body has shown that beer can have a beneficial effect, including the strengthening of the immune system and be a source of vitamins and minerals. Promoting beer as a nutritional element of the diet may create—especially among young people—the belief that there are no health consequences associated with beer consumption, which—considered a beneficial element of the diet—will be an alcoholic drink consumed in large quantities. Thus, many researchers emphasize that such benefits came only from consuming beer in very limited amounts. Moderate alcohol consumption, defined as up to one drink (12 g of ethanol) daily for women and up to two for men, seems to have beneficial effects on general health. Because of that, the increasing demand for functional beers is promising and could be recommended only when no eating disorders or problems with alcohol occur. “Functional” craft beer is brewed according to ancient and old practices, classic technologies and often with the addition of herbs, fruits and spices, which can affect the sensory and nutritional qualities of the beer. 

### 4.5. Limitations of the Study

There were many positive values of this research. On the other hand, there were also some limitations. The sample size (mainly because of the age range taken into account in this study) was rather small to make strong conclusions. After adjusting to the broader age range, the study population could be bigger. Furthermore, these studies lack a control group; thus, it was difficult to draw meaningful conclusions from a study and produce an erroneous result. Therefore, future studies could include more participants.

Moreover, this study presented a cross-sectional design, which could be subject to incidence-prevalence bias, also known as Neyman bias, which might influence the findings obtained from this study. 

## 5. Conclusions

The occurrence of eating disorders (EDs) in the studied population was strictly determined by gender. The conducted research has shown that women are more prone to distorted perception of their own body and compulsive disorders, and it is among females that eating disorders are more often diagnosed. In addition, on the basis of the analysis of the eating habits of the studied population group, irregularities are noticed in frequency of consumption of selected groups of food products, which is one of the main reasons for the initiation of eating disorders. This study opens a new field that will help health care professionals recognize the problems with eating disorders and treat them based on gender.

## Figures and Tables

**Table 1 nutrients-13-02819-t001:** The characteristics of the population studied and the number of questionnaires (NQ) analyzed.

Variable	Males (*n* = 48) MEAN ± SD	Females (*n* = 161) MEAN ± SD	*p*
NQ excluded	21	13	
NQ analyzed	27	148	
Age [years]	21.9 ± 2.00	21.3 ± 1.80	0.0882
Height [m]	1.8 ± 1.29	1.7 ± 1.36	0.0009
Weight [kg]	75.2 ± 1.86	60.8 ± 1.45	0.0008
BMI [kg/m^2^]	23.7 ± 3.30	22.0 ± 3.50	0.0190

**Table 2 nutrients-13-02819-t002:** ESP questionnaire [17].

Question	Response	
Are you satisfied with your eating patterns? (A “no” to this question was classified as an atypical response).	Yes	No
Do you ever eat in secret? (A “yes” to this and all other questions was classified as an atypical response).	Yes	No
Does your weight affect the way you feel about yourself?	Yes	No
Have any members of your family suffered from an eating disorder?	Yes	No
Do you currently suffer or have you ever suffered in the past from an eating disorder?	Yes	No

**Table 3 nutrients-13-02819-t003:** Sex differences according to the body mass index (BMI) in studied population.

BMI Category	Males		Females		All	
	*n*	%	*n*	%	*n*	%
Severe thinness	1	3.7	0	0	1	0.6
Moderate thinness	1	3.7	1	0.7	2	1.2
Mild thinness	0	0	23	15.6	23	13.1
Normal	15	55.6	96	64.9	111	63.4
Overweight	10	37.0	21	14.2	31	17.7
Obese	0	0	7	4.7	7	4.0
All population	27	100	148	100	175	100

The BMI of all respondents ranged from 14.7 kg/m^2^ to 33.5 kg/m^2^. These values included all BMI categories, from severe thinness to class I obesity. The body mass index ranged from 14.7 kg/m^2^ to 30.3 kg/m^2^ (mean ± SD: 23.7 ± 3.30) in males and from 16.8 kg/m^2^ to 33.6 kg/m^2^ (mean ± SD: 22.0 ± 3.47) in females.

**Table 4 nutrients-13-02819-t004:** The differences in the eating disorder screen for primary care (ESP) results between males and females.

Question	Y		N		*p*
	Males	Females	Males	Females	
Are you satisfied with your eating patterns?	70.4% *n* = 19	39.2% *n* = 58	29.6% *n* = 8	60.8% *n* = 90	**0.0027**
Do you ever eat in secret?	14.8% *n* = 4	32.4% *n* = 48	85.2% *n* = 23	67.6% *n* = 100	**0.0027**
Does your weight affect the way you feel about yourself?	12.8% *n* = 17	87.2% *n* = 116	23.8% *n* = 10	76.2% *n* = 32	0.0846
Have any members of your family suffered from an eating disorder?	33.3% *n* = 11	66.7% *n* = 22	11.3% *n* = 16	88.7% *n* = 126	**0.0016**
Do you currently suffer or have you ever suffered in the past from an eating disorder?	11.10% *n* = 3	21.6% *n* = 32	88.9% *n* = 24	78.3% *n* = 116	0.2092

Note; Y—yes; N—no; *p*—probability; *p* < 0.05 were indicated by using bold.

**Table 5 nutrients-13-02819-t005:** Answers to questions on habitual overeating in males and females.

Answers to Questions	Habitual Overeating						*p*
	Total		Males		Females		
	*n* = 175	%	*n* = 27	%	*n* = 148	%	
I often think about eating	102	58.3	16	59.3	86	58.1	0.9112
Eating in an important part of my life	139	79.4	20	74.1	119	80.4	0.4542
Sometimes I snack in secret even when not hungry	33	18.9	2	7.4	31	20.9	0.0982
I eat often even when I feel full	68	38.9	7	25.9	61	41.2	0.1339
I rarely overeat *	57	32.6	3	11.1	54	36.5	**0.0097**
I eat often even when not hungry	66	37.7	11	40.7	55	37.2	0.7242
Sometimes I eat uninhibited	70	40.0	13	48.1	57	38.5	0.3613
I rarely feel overeaten *	73	41.7	6	22.2	67	45.3	**0.0255**
Food is too important to me	65	37.1	7	25.9	58	39.2	0.1896
My stomach is like a bottomless pit	37	21.1	5	18.5	32	21.6	0.7165

Note: * The question is scored for “NO” answer; *p*—probability; *p* < 0.05 were indicated by using bold.

**Table 6 nutrients-13-02819-t006:** Answers to questions on emotional overeating in males and females.

Answers to Questions	Emotional Overeating						*p*
	Total		Males		Females		
	*n* = 175	%	*n* = 27	%	*n* = 148	%	
I often feel guilty when I eat too much	80	45.7	4	14.8	76	51.4	**0.0005**
Other people comment on my eating habits	57	32.6	13	48.1	44	29.7	0.0604
I eat more than normal when I am upset	73	41.7	9	33.3	64	43.2	0.3369
Sometimes when I start to eat, I feel that I won’t be able to tell myself “enough”	48	27.4	3	11.1	45	30.4	**0.0388**
I would prefer to weigh less than I do now	118	67.4	14	51.9	104	70.3	0.0604
My diet generally depends on my mood	110	62.9	13	48.1	97	65.5	0.0854
In the company, I eat less but “allow” myself more when I am alone	46	26.3	4	14.8	42	28.4	0.1409
When I get angry I start eating	52	29.7	46	170.4	6	4.1	0.3543
I feel like I need to do some physical activities after having a large meal	76	43.4	10	37.0	66	44.6	0.4663
Food puts me in a good mood	143	81.7	24	88.9	119	80.4	0.2943

Abbreviations as in Table 4; *p* < 0.05 were indicated by using bold.

**Table 7 nutrients-13-02819-t007:** The differences in the emotional overeating habits between males and females.

Answers to Questions	Y		N		*p*
	Males	Females	Males	Females	
I often feel guilty when I eat too much	14.8% *n* = 4	51.4% *n* = 76	85.2% *n* = 23	54.7% *n* = 81	**0.0005**
Sometimes when I start to eat, I feel that I won’t be able to stop	11.1% *n* = 3	30.4% *n* = 45	88.9% *n* = 24	69.6% *n* = 103	**0.0388**
My eating habits generally depend on my mood	48.2% *n* = 13	65.5% *n* = 97	51.9% *n* = 14	34.5% *n* = 51	0.0854

Abbreviations as in Table 4; *p* < 0.05 were indicated by using bold.

**Table 8 nutrients-13-02819-t008:** Answers to questions on dietary restrictions in males and females.

Answers to Questions	Dietary Restrictions						*p*
	Total		Males		Females		
	*n* = 175	%	*n* = 27	%	*n* = 148	%	
I rarely follow diets *	30	17.1	7	25.9	23	15.5	0.1879
I read and collect diets from magazines and books	16	9.1	2	7.4	14	9.5	0.7337
I rarely worry about my weight	78	44.6	14	51.9	64	43.2	0.4079
I am not satisfied with my body	110	62.9	14	51.9	96	64.9	0.1981
Sometimes, I avoid eating even when I’m hungry	42	24.0	3	11.1	39	26.4	0.0882
I put too much weight	69	39.4	6	22.2	63	42.6	**0.0467**
I rarely feel guilty after overeating *	81	46.3	6	22.2	75	50.7	**0.0064**
I like to feel empty in my stomach	19	10.9	3	11.1	16	10.8	0.9632
I often follow diets	19	10.9	5	18.5	14	9.5	0.1641
I consciously limit my food consumption	51	29.1	10	37.0	41	27.7	0.3372

Note: * The question is scored for “NO” answer; *p*—probability; *p* < 0.05 were indicated by using bold.

**Table 9 nutrients-13-02819-t009:** The differences in dietary restrictions results between males and females.

Statement	Y		N		*p*
	Males	Females	Males	Females	
I pay too much attention to my weight	22.2% *n* = 6	42.6% *n* = 63	77.8% *n* = 21	57.4% *n* = 85	**0.0466**
I rarely feel guilty after overeating	77.8% *n* = 21	49.3% *n* = 73	22.2% *n* = 6	50.7% *n* = 75	**0.0070**
Sometimes I avoid eating even when I’m hungry	11.1% *n* = 3	26.4% *n* = 39	88.9% *n* = 24	73.6% *n* = 109	0.0882

*p*—probability; *p* < 0.05 were indicated by using bold.

**Table 10 nutrients-13-02819-t010:** The mean and standard deviation (SD) values for different classes of eating habits in males and females.

Factor	Total		Males		Females		*p*
	Mean	SD	Mean	SD	Mean	SD	
Habitual overeating	4.1	3.03	3.3	2.37	4.2	3.13	0.331
Emotional overeating	4.6	2.57	3.7	2.30	4.8	2.59	**0.044**
Dietary restrictions	3.1	2.32	2.6	2.04	3.1	2.36	0.301

Abbreviations as in Table 4; *p* < 0.05 were indicated by using bold.

**Table 11 nutrients-13-02819-t011:** The distribution of responses selected from the “My Eating Habits” questionnaire in accordance with behaviors typical for people with anorexia nervosa in males.

ESP vs. MEH Questions and Statements	Do you Currently Suffer or Have You Ever Suffered in the Past from an Eating Disorder? (ESP Question)		*p*
I am not satisfied with my body image	YES	NO	0.5959
YES	66.7% *n* = 2	50.0% *n* = 12	
NO	33.3% *n* = 1	50.0% *n* = 12	
I’d rather weigh less than I do now	YES	NO	0.4959
YES	33.3% *n* = 1	54.2% *n* = 13	
NO	66.7% *n* = 2	45.8% *n* = 11	
I prefer to prepare meals myself	YES	NO	0.8815
YES	66.7% *n* = 2	70.8% *n* = 17	
NO	33.3% *n* = 1	29.2% *n* = 7	
Food matters too much to me	YES	NO	0.7562
YES	33.3% *n* = 1	25.0% *n* = 8	
NO	66.7% *n* = 2	75.0% *n* = 18	
After a larger meal, I want to get rid of excessed calories.	YES	NO	0.8879
YES	33.3% *n* = 1	37.5% *n* = 9	
NO	66.7% *n* = 2	62.5% *n* = 15	

Abbreviations as in Table 4.

**Table 12 nutrients-13-02819-t012:** The distribution of responses to selected statements from the “My Eating Habits” questionnaire in accordance with behaviors typical for people with anorexia nervosa in females.

ESP vs. MEH Questions and Statements	Do you Currently Suffer or Have You Ever Suffered in the Past from an Eating Disorder? (ESP Question)		*p*
I am not satisfied with my body	YES	NO	**0.0024**
YES	85.7% *n* = 28	58.6% *n* = 68	
NO	12.5% *n* = 4	41.4% *n* = 48	
I’d rather weigh less than I do now	YES	NO	**0.0160**
YES	87.5% *n* = 28	65.5% *n* = 76	
NO	12.5% *n* = 4	34.5% *n* = 40	
I prefer to prepare meals myself	YES	NO	**0.0031**
YES	87.5% *n* = 28	59.5% *n* = 69	
NO	12.5% *n* = 4	40.5% *n* = 47	
Food matters too much to me	YES	NO	**0.0005**
YES	65.6% *n* = 21	31.9% *n* = 37	
NO	34.4% *n* = 11	69.1% *n* = 79	
After a larger meal. I want to get rid of excessed calories	YES	NO	**0.0001**
YES	75.0% *n* = 24	36.2% *n* = 42	
NO	25.0% *n* = 8	63.8% *n* = 74	

Abbreviations as in Table 4; *p* < 0.05 were indicated by using bold.

**Table 13 nutrients-13-02819-t013:** The distribution of responses to selected statements from the “My Eating Habits” questionnaire in accordance with behaviors typical for people with binge-eating disorder in males.

ESP vs. MEH Questions and Statements	Do you Currently Suffer or Have You Ever Suffered in the Past from an Eating Disorder? (ESP Question)		*p*
I often eat when I am not hungry	YES	NO	0.2332
YES	0% *n* = 0	33.3% *n* = 8	
NO	100% *n* = 3	66.7% *n* = 16	
Sometimes I stuff myself with food	YES	NO	0.5859
YES	33.3% *n* = 1	50.0% *n* = 12	
NO	66.7% *n* = 2	50.0% *n* = 12	
Sometimes, when I eat too much. I feel guilty	YES	NO	0.3382
YES	33.3% *n* = 1	12.5% *n* = 3	
NO	66.7% *n* = 2	87.5% *n* = 21	

Abbreviations as in Table 4.

**Table 14 nutrients-13-02819-t014:** The distribution of responses to selected statements from the My Eating Habits questionnaire in accordance with behaviors typical for people with compulsive overeating syndrome in females.

ESP vs. MEH Questions and Statements	Do you Currently Suffer or Have You Ever Suffered in the Past from an Eating Disorder? (ESP Question)		*p*
I often eat when I am not hungry	YES	NO	**0.0047**
YES	56.3% *n* = 18	29.3% *n* = 34	
NO	43.8% *n* = 14	70.7% *n* = 82	
Sometimes I stuff myself with food	YES	NO	**0.0218**
YES	56.3% *n* = 18	33.9% *n* = 39	
NO	43.8% *n* = 14	66.1% *n* = 76	
Sometimes, when I eat too much. I feel guilty	YES	NO	**0.00002**
YES	84.4% *n* = 27	42.2% *n* = 49	
NO	15.6% *n* = 5	57.8% *n* = 67	

Abbreviations as in Table 4; *p* < 0.05 were indicated by using bold.

**Table 15 nutrients-13-02819-t015:** The distribution of responses selected from the “My Eating Habits” questionnaire in accordance with the satisfaction with eating patterns in males.

ESP vs. MEH Questions and Statements	Are You Satisfied with Your Eating Patterns? (ESP Question)		*p*
I often skip meals	YES	NO	0.56114
YES	26.3% *n* = 5	37.5% *n* = 3	
NO	73.7% *n* = 14	62.5% *n* = 5	
I generally eat only 3 meals a day	YES	NO	0.73246
YES	31.6% *n* = 6	25.0% *n* = 2	
NO	68.4% *n* = 13	75.0% *n* = 6	
I rarely eat meals at strictly scheduled times	YES	NO	0.7657
YES	68.4% *n* = 13	62.5% *n* = 5	
NO	31.6% *n* = 6	37.5% *n* = 3	

Abbreviations as in Table 4.

**Table 16 nutrients-13-02819-t016:** The distribution of responses selected from the “My Eating Habits” questionnaire in accordance with the satisfaction with eating patterns in females.

ESP vs. MEH Questions and Statements	Are You Satisfied with Your Eating Patterns? (ESP Question)		*p*
I often skip meals	YES	NO	**0.00147**
YES	31.0% *n* = 18	57.8% *n* = 52	
NO	69.0% *n* = 40	42.2% *n* = 38	
I generally eat only 3 meals a day	YES	NO	0.6636
YES	22.4% *n* = 13	25.6% *n* = 23	
NO	77.6% *n* = 45	74.4% *n* = 67	
I rarely eat meals at strictly scheduled times	YES	NO	**0.0106**
YES	74.1% *n* = 43	90.0% *n* = 81	
NO	25.9% *n* = 15	10.0% *n* = 9	

Abbreviations as in Table 4; *p* < 0.05 were indicated by using bold.

**Table 17 nutrients-13-02819-t017:** Kind and frequency (%; *n*) of different food products consumption in males and females.

Food Items	(1)	(2)	(3)	(4)	(5)	(6)	(7)	(8)	(9)	(10)	*p*
**Fruits, all kind**											
*Males*		3.7; 1	11.1; 1	7.4; 2	14.8; 4	11.1; 3	18.5; 5	11.1; 3	11.1; 3	11.1; 3	0.5107
*Females*	2; 3	0.7; 1	4.7; 7	17.6; 26	21; 31	13.5; 20	17.6; 26	11.5; 17	4.7; 7	6.8; 10	
**Vegetables, all kind**											
*Males*	-	-	7.4; 2	7.4; 2	14.8; 4	7.4; 2	22.2; 6	22.2; 6	11.1; 3	7.4; 2	
*Females*	-	1.4; 2	5.4; 8	9.5; 14	14.8; 22	17.6; 26	21.6; 22	13.5; 20	8.8; 13	7.4; 11	0.9062
**Dry pulses**											
*Males*	14.8; 4	22.2; 6	14.8; 4	22.2; 6	7.4; 2	7.4; 2	3.7; 1	3.7; 1	3.7; 1	-	
*Females*	23.7; 35	26.4; 39	23; 34	11.5; 17	8.8; 13	2.7; 4	2.7; 4	-	-	1.4	0.0546
**Fresh pulses and canned pulses**											
*Males*	7.4; 2	18.5; 5	22.2; 6	14.8; 4	7.4; 2	14.8; 4	11.1; 3	-	-	3.7; 1	0.0112
*Females*	11.5; 17	22.3; 33	25; 37	17.6; 26	14.2; 21	6.8; 10	1.4; 2	-	-	1.4; 2	
**Red meat**											
*Males*	11.1; 3	14.8; 4	3.7; 1	18.5; 5	25.9; 7	3.7; 1	18.5; 5	3.7; 1	-	-	**0.0025**
*Females*	21; 31	22.3; 33	21.6; 32	12.8; 19	14.2; 21	4.7; 7	2; 3	0.7; 1	-	0.7; 1	
**Poultry meat**											
*Males*	-	7.4; 2	3.7; 1	7.4; 2	22.2; 6	18.5; 5	22.2; 6	14.8; 4	-	3.7; 1	0.1115
*Females*	5.4; 8	7.4; 11	10.8; 16	22.3; 33	23.7; 35	12.2; 18	9.5; 14	4.7; 7	2.7; 4	1.4; 2	
**Lean fish**											
*Males*	3.7; 1	29.6; 8	3.7; 1	40.7; 11	11.1; 3	3.7; 1	3.7; 1	-	3.7; 11	-	**0.0316**
*Females*	17.7; 26	25.2; 37	22.5; 33	20.4; 30	5.4; 8	3.4; 5	3.4; 5	1.4; 2	-	0.7; 1	
**Fatty fish**											
*Males*	18.5; 5	22.2; 6	14.8; 4	29.6; 8	3.7; 1	-	-	7.4; 2	3.7; 1	-	0.1547
*Females*	22.3; 33	25.7; 38	24.3; 36	15.5; 23	5.4; 8	2; 3	2; 3	0.7; 1	0.7; 1	1.4; 2	
**Milk and natural milky drinks**											
*Males*	3.7; 1	-	7.4; 2	18.5; 5	14.8; 4	7.4; 2	22.2; 6	18.5; 5	7.4; 2	-	0.8244
*Females*	3.4; 5	6.1; 9	7.4; 11	17.6; 26	19.6; 29	8.1; 12	18.2; 27	10.8; 16	4.1; 6	4.7; 7	
**Cheeses**											
*Males*	3.7; 1	-	11.1; 3	18.5; 5	22.2; 6	3.7; 1	25.9; 7	7.4; 2	7.4; 2	-	0.5107
*Females*	4.7; 7	10.1; 15	14.2; 21	12.8, 19	23.7; 35	14.9; 22	9.5; 14	6.1; 9	2; 3	2; 3	
**Beer**											
*Males*	11.1; 3	11.1; 3	11.1; 3	7.4; 2	18.5; 5	22.2; 6	3.7; 1	7.4; 2	7.4; 2	-	
*Females*	12.6; 18	24.3; 36	27.7; 41	19.6; 29	6.1; 9	4.7; 7	2.7; 4	2; 3	0.7; 1	-	**0.00006**
**Vodka and strong liquor**											
*Males*	22.2; 6	22.2; 6	25.9; 7	3.7; 1	3.7; 1	14.8; 4	-	3.7; 1	-	3.7; 1	
*Females*	27.4; 40	34.5; 51	16.9; 25	12.8; 19	4.1; 6	0.7; 1	2; 3	0.7; 1	0.7; 1	0.7; 1	**0.00383**

Explanation: (1) never; (2) once a month or less; (3) 2–3 times a week; (4) 1–2 times a week; (5) 3–5 times a week; (6) 5–6 times a week; (7) once daily; (8) 2–3 times a day; (9) 4–5 times a day; (10) 6 times a day or more; *p* < 0.05 were indicated by using bold.

## Data Availability

The data presented in this study are available on reasonable request from the corresponding author.

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
