# Peer review of "Risk Factors for Eating Disorders and Perception of Body in Young Adults Associated with Sex"

_nutrients, 2021, doi:10.3390/nu13082819_

Round 1

Reviewer 1 Report

The manuscript presents and interesting study, of high importance for the scientific community. But  I have included a few suggestions that could further clarify the manuscript:

L70-71: The aim of this study was to examine the differences in food choices and eating behaviorbetween males and females. - Where is the nolvely and why now? Please explain.

L493-497: which consists in limiting the consumption of food during the day in favor of alcohol consumption in the evening hours. - Hardly to believe such statement, if refer to alcohol-addiction, than it is the case.

Drunkorexia: A term coined to refer to condition of binge drinking combined with the typical self-imposed starvation seen with anorexia nervosa. It has also been used to refer to individuals who use purging (as seen with bulimia nervosa) to try to reduce caloric intake to offset the calories consumed in alcohol. And this is a special group. But L485-492 is describing a pretty normal life situation,a nyway where is the link to the study and the link between the 2 described situations related to alcohol.

Besides, beer is considered as a nutritious and refreshing carbonated beverage, which is why its production is spread all over the world, rich in different minerals and vitamins. Today’s research is focused on improving the functionality of different beers and beer-based beverages. Therefore, media has a high influence on consumption, therefore these aspects maybe have to be discussed.

The increasing demand for functional beers is promising, and how to explain to a person  suffering of drunkorexia to limit the consuption?!

L507-509: This study opens new field,which will help health care professionals recognize the problems with eating disorders and treat them based on gender. - How? Please explain, this is only a hypothesis.

Reviewer 2 Report

This is a cross-sectional study including 148 females and 27 males’ participants. Various Eating Disorder Screen characteristics were recorded and compared among females and males. Stratifications by the ESP results were made regarding different gender characteristic. They found that sex (females) was a risk factor for Eating Disorders and perception of body in young adults.

This is an interesting study with some new findings in this area of research. The sample size of subjects is small for analysis. However, I nevertheless have the following comments that required to be addressed.

  1. The study design should be specified in this study. The authors should clarify this concern.
  2. The statistical methods used and described very well. Why use Duncan’s test for post hoc analysis?
  3. How does the authors to determine the sample size of this study? Please use power analysis to statement adequate sample size in this study.
  4. For tables, I suggested to move some tables as supplementary, such as Table 2.
  5. We know the females as a risk factor for Eating Disorder. How is the odds ratio (or relative risk) refer to males? The authors should clarify this concern.
  6. It seems the gender as a common risk factor discussed in previous study. The authors should highlight novelty in this study. What do we newly learned from this study?
  7. Lastly, the authors only briefly discuss limitations, but didn’t acknowledging that the one of limitation include the cross-sectional design. They should elaborate on how the use of this design is subject to Incidence-Prevalence bias, also known as Neyman bias, and how that might influence their findings.

Reviewer 3 Report

Comments to Authors 

            This study showed that women are more prone to distorted perception of their own body and compulsive disorders, and it is among the female that eating disorders are more often diagnosed.

           Eating disorders are caused by a variety of factors that are unclear. Many experts, on the other hand, feel that a mix of genetic, physical, social, and psychological variables may play a role in the onset of an eating disorder. Eating disorders can also be influenced by societal pressure. In Western culture, physical attractiveness and a thin physique are frequently associated with success and personal value. Eating disorder behaviors may be fueled by a desire to succeed or be accepted. By investigating child, parental and teaching interventions and including outcomes such as weight control and disordered eating behaviors, a trend toward a reduction in eating disorder risk factors was observed, particularly body image-related outomes in girls [1]. Negative body feelings and reduced body protection fully mediated the relationship between self-esteem and binge eating, regardless of gender [2].

            Authors are kindly requested to emphasize the current concepts about these issues in the context of recent knowledge and the available literature. This articles should be quoted in the References list.

References

  1. Disordered eating, body image concerns, and weight control behaviors in primary school aged children: A systematic review and meta-analysis of universal-selective prevention interventions [published online ahead of print, 2021 Jul 10]. Int J Eat Disord. 2021;10.1002/eat.23571. doi:10.1002/eat.23571.
  2. Self-Esteem and Binge Eating among Adolescent Boys and Girls: The Role of Body Disinvestment. Int J Environ Res Public Health. 2021;18(14):7496. Published 2021 Jul 14. doi:10.3390/ijerph18147496.

Round 2

Reviewer 1 Report

The authors have clearly improved the manuscript according to the suggestions and made it really clearer. It is suitable for publication.

Reviewer 2 Report

Thanks for your efforts on revision. Please add Neyman bias as a limitation in this study.
